# Entropy-Based Behavioural Efficiency of the Financial Market

**DOI:** 10.3390/e23111396

**Published:** 2021-10-24

**Authors:** Emil Dinga, Camelia Oprean-Stan, Cristina-Roxana Tănăsescu, Vasile Brătian, Gabriela-Mariana Ionescu

**Affiliations:** 1Center for Financial and Monetary Research, Romanian Academy, “Victor Slăvescu”, 050711 Bucharest, Romania; emildinga2004@gmail.com; 2Faculty of Economic Sciences, Lucian Blaga University of Sibiu, 550324 Sibiu, Romania; camelia.oprean@ulbsibiu.ro (C.O.-S.); cristina.tanasescu@ulbsibiu.ro (C.-R.T.); vasile.bratian@ulbsibiu.ro (V.B.); 3School of Advanced Studies of the Romanian Academy (SCOSAAR), Romanian Academy, 010071 Bucharest, Romania

**Keywords:** behaviour, entropy, efficiency, implicit information, financial market, EMH, AMH, EBBE

## Abstract

The most known and used abstract model of the financial market is based on the concept of the informational efficiency (EMH) of that market. The paper proposes an alternative which could be named the behavioural efficiency of the financial market, which is based on the behavioural entropy instead of the informational entropy. More specifically, the paper supports the idea that, in the financial market, the only measure (if any) of the entropy is the available behaviours indicated by the implicit information. Therefore, the behavioural entropy is linked to the concept of behavioural efficiency. The paper argues that, in fact, in the financial markets, there is not a (real) informational efficiency, but there exists a behavioural efficiency instead. The proposal is based both on a new typology of information in the financial market (which provides the concept of implicit information—that is, that information ”translated” by the economic agents from observing the actual behaviours) and on a non-linear (more exactly, a logistic) curve linking the behavioural entropy to the behavioural efficiency of the financial markets. Finally, the paper proposes a synergic overcoming of both EMH and AMH based on the new concept of behavioural entropy in the financial market.

## 1. Introduction

The basic objective of this paper is to put into discussion a new concept of market efficiency, namely the entropy-based behavioural efficiency (EBBE). As denomination suggests, there are three concepts involved here: (a) the entropy—which will be defined in its own signification on the financial market (very different from the three well-known concepts in literature: thermodynamic, statistical, and informational entropy, respectively); (b) the behaviour—which will be invested with the quality of driver on the financial market—differently from the current adjudication of the information as driver; (c) the efficiency—which will still hold its role of impersonal target of the financial market functioning. The aim to which the paper aspires is therefore to deliver, for the scientific community debates, the proposal of replacing the informational entropy with the behavioural entropy, and, respectively, of replacing the informational efficiency with the behavioural efficiency in the financial market functioning. The methodology used in the paper is that of a logical approach, in order to avoid the (too intensive) context conditioning which, at least in principle, can reduce the generality (that is, the factual testability chances) of the findings. Of course, the most significant results which behaviourism and evolutionism already provide to analysts are taken into account but, also, they are treated in an abstract way, as inputs from biology, psychology, culture, axiology, and normativity areas. More specifically, the methodological approach of the paper holds the following predicates: (a) it is rather abductive—for example, as a response to the (recognized, in the literature) impossibility to empirically test the informational efficiency of the financial market, we have formulated the most plausible ”explanation”: this impossibility is generated by the fact that, in the financial market, there is a primacy of the behaviour over the information and, from this point further, we have proceeded deductively; (b) it is rather much de-contextualized from the empirical (current) financial markets, in order to capture the most general and abstract features of financial markets functioning; (c) it is permanently oriented to the evolutionary paradigm of the financial market. Authors strongly believe that the evolutionarism (including institutionalism) is the necessary road of modelling, formalizing, predicting, and developing the financial markets theory; (d) from a conceptual perspective, the paper adjudications the presupposition that the entropy is nothing else than a measure of order—of course, that order must be defined in a specific way so it is appropriate to the financial markets and to the research interest.

The research is organized as follows: a very short re-bringing to mind of the currently used concepts of entropy, without any technicalities which would not serve further the own research presented. Section State of the art presents a status quo regarding the main positions related to the concept of entropy, including the economic/financial field. Section Background provides, firstly, an opinion regarding the typology of information on the financial market, and the mechanism involved, in order to separate a crucial species—implicit information—which will be massively used further as the driver of behaviour, and secondly, exposes a point of view regarding the concept of behavioural efficiency—as opposed to that of informational efficiency. Section Proposal introduces the concepts of behavioural entropy, behavioural efficiency, and then, the concept of entropy-based behavioural efficiency—the key concept of our paper. The next section, Discussion, put into debate seven questions that have arisen around the proposal made, including suggestions to measure the behavioural entropy (as opposed to informational entropy), the behavioural efficiency and ways in which the predictions based on the proposal can be empirically tested (by the Popperian falsifiability). Section Results synthesizes the main findings of research, and Section Conclusions draws the most relevant significations of the research in relation with the financial theory, methodology, and instrumentalization. Some directions of further directions/objectives of research which seem be suggested by the proposal are finally drawn.

## 2. Briefly on Currently Used Concepts of Entropy

Essentially, the concept of entropy is linked to the more general concept of the order. For a generic observer, an order of an entity (either static or dynamic) is simply a coincidence (over a given significance threshold) between the perceived configuration of that entity and an already existent configuration in a list of possible configurations of that entity. If such a coincidence does not happen, no order is observed (Nota bene: of course, the observer can, in such a case, decree that a new order has arisen in the world and, consequently, that new configuration is recorded in her/his list of orders for further use). According to this definition of the concept of order, it results that the concept of disorder, meaning something as opposite to order, lacks any sense. In this context, the concept of entropy is viewed as a measure (qualitative and even quantitative) of the order. We would even say that the entropy is a measure of the degree of the coincidence between the perceived order and the “twin” listed order in the perceiver’s mind. A first logical consequence of such a positioning is the following: the entropy is not at all a pure objective property of an entity (system, event, process, and so on), but (like any cognizance) it is a relationship between objectivity and subjectivity. A second logical consequence consists in the fact that the lowest degree of order always implies the homogeneity (at least over a perceptible degree/threshold) of the entity in case, under the criterion of interest: temperature, structure, information, behaviour, etc. By (contingent) convention, the entropy measure is set such as it moves inversely proportional with the degree of order—that is, the maximum of order is equivalent to the maximum of heterogeneity, while the maximum of entropy or the minimum of order is equivalent to the maximum of homogeneity. Shortly, as said above, the entropy is a measure of the (subjectively selected) order.

Generally (and especially for Entropy’s readers), the concept of entropy is well-known. Consequently, here only a few considerations—focused on the essence of this concept—will be provided in preparing our own concept of entropy specified for the financial market.

The term entropy (from Greek: εντροπία, formed from "εν"—to, and "τροπή"—turning) means to go to…, to turn into the direction. The meaning is that of a necessary propensity of a system/process/phenomenon in an unambiguous direction.

The main (orthodox) predicates of the concept of entropy seems to be:It is a state-function, not a process-function. Consequently, the value of the entropy variation does not depend on the intermediate stages ("road"), but only on the initial and final points (Nota bene: dependence on intermediate stages leads to process-functions).It is a macroscopic value (see Boltzmann’s relation for entropy): more precisely, it signifies a macroscopic irreversibility derived from a microscopic reversibility (see, here, also the problem of Maxwell’s demon).It is a statistical quantity (based on the statistical formulation of Thermodynamics); this justifies the occurrence of probability in the analytical formula of entropy in statistical Thermodynamics (because probabilities can only model the average of a population) (Nota bene: in reality, Boltzmann does not consider probabilities in their usual sense, i.e., inductive derivatives, as is the case, for example, of objective probabilities, but rather as possibilities; by possibilities we mean states or events, necessary or contingent, unrelated to a previous state archive—in such a context, the concept of propensity, initiated by Karl Popper following Aristotle’s Physics seems to us much more adequate).It is an additive value.

There are three distinct types of the concept of entropy [1]:i)Phenomenological entropy—a measure of the macroscopic entropy based on Thermodynamics, that is, anchored in macroscopic properties as heat and temperature) (initiated by Clausius, 1865):$dS=\frac{dQ}{T}$, where *S* is the entropy, *T* is the absolute (non-empirical) temperature. Signification is: the measure of thermal energy that cannot be transformed into mechanical work; to be noted that the phenomenological entropy is of ontological type.ii)Statistical entropy—based on a measure of macroscopic aggregation of microscopic states (initiated by Boltzmann, 1870):$S=k\cdot \ln\left(\Omega \right)$ where: k is the Boltzmann constant and Ω is the total number of microstates of the analyzed microstate. Signification is: the measure of the distribution of microscopic states in a macroscopic system. In 1876, Gibbs introduces his own concept of entropy, which is developed, in 1927, by von Neumann as von Neumann entropy.iii)Informational entropy—a measure of entropy based on the probability of states (initiated by Shannon, 1948). In fact, Shannon introduces his concept of informational entropy based on considerations of uncertainty, being a remake of Boltzmann’s entropy in a form which includes the uncertainty. Nota bene: the probability is involved both in the statistical entropy and in informational entropy, but with a notable difference: statistical entropy uses the objective non-frequential probability, known especially as propensity [2], while the informational entropy uses rather frequential probability, that is, a probability drawn from an archive of the given experiments of interest (for example, for verbal lexicon processes, see Shannon informational entropy):$S\left(X\right)=-\sum_{i\;=1}^{n}p\left(x_{i}\right)\cdot \log_{b}p\left(x_{i}\right),$ where X is a discrete variable $(X\left(x_{1},x_{2},\ldots ,x_{n}\right)$, and p is a probability function (generally, b=2, which gives information measured as bits). For the continuous case, $S\left(X\right)=-\int_{-\infty }^{\infty }f\left(x\right)\cdot \log_{b}f\left(x\right)dx$, where X is a continuous variable, with the distribution function $f\left(x\right)$. Signification: the measure of uncertainty associated with a random variable (also indicates the amount of information contained in a message, or the minimum length of the message to communicate information). To be mentioned is that, in 1988, Tsallis generalized Boltzmann’s entropy as Tsallis’s entropy.

Alternatives of the concept of entropy have been done for specific fields: for example, for the Quantum Theory, von Neumann (1927) provided the expression: $S=-tr\left[\rho \cdot \ln\left(\rho \right)\right]$, where ρ the density matrix, and tr is the trace of the density matrix. Signification: by writing the density matrix in terms of its eigenvalues, Shannon’s formula is obtained.

From a purely mathematical perspective, a larger list of different categories of entropy (of course, exclusively as informational entropies), including the relationships among them is provided in [3].

In our opinion, the concept of the entropy could be particularized especially for the social/economic field, based on the following suggestions:In economic field: as a measure of free energy (not related to an energy stock) in a given system (i.e., a measure of the energetic disorder);In social field: as a measure of anomie (i.e., of the normative disorder) [4];In (scientific) knowledge field: as a measure of non-explanatory coverage (i.e., of the causal disorder). Nota bene: the link with Kuhn’s concept of paradigm is, here, unavoidable;In art field: as a measure of meaning non-coverage (i.e., of a meaning disorder); Nota bene: for example, the current Post-Modernism.

Regarding the economic field, we want to provide a short mention on the entropic model of the economic process, initiated by Nicholas Georgescu-Roegen [5], the Romanian-American rebellious economist against the mainstream of the 1970s.

(a)The general *framework*: Georgescu-Roegen’s crucial intuition is that the economic world is not a trajectory but a process. This means the economic process is not reversible (by, for example, the simple inversion of the algebraic sign of the variable time in the economic equations) but, somewhat, it has an arrow time. Georgescu-Roegen convoked the second law of Thermodynamics—the so-called entropy law—to ground any economic process and to provide it with an arrow time, that is, a process understood as a relationship between an individual and his/her non-anthropic environment. Georgescu-Roegen called the second law of Thermodynamics the most economic law of nature (or of Physics), although, for example, in nature, there is also the principle of Maupertuis—the principle of minimum action, based on which the cosmological geodesics are built.(b)The basic *assumption*: the basic assumption of the Georgescu-Roegen entropic model of the economic process is the (inevitable) decreasing ratio between the bound energy and the free energy available for a given economic system. Although such a degradation is common to the Universe (that is a closed system, by definition), locally this assumption works also as a result of economic activity itself. Consequently, Georgescu-Roegen doubts on the real possibility to conceive and build a circular economic process—when any output reconstitutes the necessary (consumed) inputs. In this context, he makes a significant distinction between fund (an energetic reservoir without inputs, for example the Sun) and stock (an energetic reservoir with both inputs and outputs). Although Georgescu-Roegen does not support anywhere that the energy (more exactly, the bound energy) is the ultimate source of economic value—he remains into the utility theory here—many enthusiastic followers of him passed into the neo-energetics territory, which he himself severely criticized [6].(c)A comment: Georgescu-Roegen’s concept of entropy is one of the Thermodynamic species, that is, it is based on the natural propensity to homogenization (for example, regarding the ratio bound energy—free energy). In our opinion, an economic concept of entropy must be based on the behaviour of individuals, which can avoid the degradation (or, at least, can reduce the velocity of such a degradation) or, in special cases, can find (economic) utility even in the free energy. The economic entropy cannot be examined, as we understand, outside the normative framework of the society and, more than that, outside the axiological matrix of that society. We think the concept of entropy cannot be, in the general social field, more than a metaphorical one, associated (causal, structural, and functional) to a kind of order which is rather arbitrarily selected as an order of interest, scientifically or praxiologically.(d)The *heritage*: the genuine proposal of Georgescu-Roegen has not been continued at the research level with too much success (Nota bene: we think the cause is its too close link of the proposal with the ”natural” second law of Thermodynamics). Instead, two new branches (if not, in fact, one with two specificities) of Economics seems to be initiated based on that proposal; (a) Bioeconomics; (b) Ecological Economics. In our opinion, the two branches are ”illegally” grounded on the second law of Thermodynamics—they could be, logically, edified outside this hypothesis as well.(e)A “*prophecy*”: Georgescu-Roegen’s intuition can be, however, capitalized in the larger spectrum of current approaches to rebuild Economics: behaviourism, institutionalism, evolutionarism, without a too strong link to Thermodynamics. Perhaps, the behavioural-based entropy could be a key here.

## 3. State of the Art

The issue of behavioural entropy, or of behavioural efficiency on the financial market, is not discussed in the specialty literature. Much less there are debates on the putting together the two concepts, like the EBBE model in the present paper. The main reason for such a state is, in our opinion, the quasi-prejudgment (for which both the theory of martingale of Samuelson, and the theory of information efficiency—EMH—of Fama are responsible) according to which the agents search information as information in order to decide and, consequently, act/abstain. However, in reality—that is, as economic homo œconomicus, not as mathematical homo œconomicus—the agents see, observe, perceive, examine, and interpret always only behaviours. Only from behaviours the agents acquired their information—namely, our concept of implicit information. This empirical statement is sufficient (which, from a logical point of view, plays the role of a conjecture, in Popper’s sense, so it could be subject of the empirical/factual testing) we think, to “decree” a primacy of behaviour over information in financial market functioning, and, in essence, this is the “red thread” around which the paper is organized, developed, and argued. 

However, we have found such a point of view (outside the financial, economic or social field of research) that tries to replace the information determinism of evolution with the moving (more generally, the behavioural) determinism of evolution in the field of non-conscious life, more exactly, at the molecular level [7]. If such an approach is (or seems) more realistic in describing the process of non-conscious life, all the more it could be appropriate for describing the processes specific to the conscious life. In addition, even more qualified can be such an approach when this conscious life is endowed with free will and (partially) with rationality, as the financial market implies.

Also, there are, in the specialty literature some approaches of the concept of endogenously acquired information [8], but this type of information is not exactly what we propose to be the driver of the behaviour on the financial market, namely the implicit information. In other words, the exogenously acquired information is acquired through informational channels too (for example, through operating of rationality models), while our implicit information is non-informationally acquired.

The most common explanation for the lack of debates, in the speciality literature, of the concept of behavioural entropy, and of behavioural efficiency, respectively, in the financial markets, consists, of course, in the huge ”prestige” of the mathematical homo œconomicus, that is focused on information (as well as on the rationality models capable of processing that information) which discouraged (and continues to discourage) the heterodoxism in modelling the (real) economic behaviour. 

In a rather stylized way, the main observations regarding the researchers’ positioning in the matter of entropy could be synthesized as follows:(a)Based on the informational entropy formalizations (especially of Boltzmann and Shannon) most approaches of the concept of entropy are kept in the limits of the informational entropy [9].This state allows a large manoeuvre room from the instrumental point of view but ignores the behaviour—for example, a portfolio of assets is optimized by entropy per se, simply by replacing Markowitz’s variance method with one based on entropy [9].(b)The few positions linked to the Thermodynamic entropy (as ontological entropy) fall into energetism, putting at the basis of economic value the (bound) energy, often as an alternative to Marxist theory of value.(c)At least for the economic/financial field, it is (at large) maintained the position according to which the entropy must be connected to the uncertainty (from here, by ignoring the venerable warning of Knight, the entropy is easy linked to risk).(d)Again ”inspired” by the concept of informational entropy, the probability (including the frequential one) is omnipresent in the entropy formalization (Nota bene: in our opinion, the probability could be replaced in two alternative ways: (a) by using the Bayes probability, which ”recapture” the behaviour; (b) by constructing the propensity—as, in fact, proceeded Boltzmann).

## 4. Background

### 4.1. Information vs. Behaviour in the Financial Market

In the last instance, all the above mentioned three kinds of entropy are originated in information (in our opinion, information is born whenever a sign—in the semiotic sense—removes an indeterminacy in some a degree; therefore, information can flow even if no human observer is involved into the transmission of that sign; moreover, the presence of a human observer is required only to transform, if is the case, the information into cognizance and then, into knowledge). All the notorious theories of the financial market (beginning with Fama’s Efficient Market Hypothesis), including the less recent behaviourism and the more recent Lo’s Adaptive Market Hypothesis [10] consider that information drives the behaviour. Consequently, they build their basic assumptions and, of course, mathematical models, on this principle. However, in our opinion, such a principle—the primacy of information over the behaviour—is not at all out of doubt. To support such an assertion, we shall present some considerations on the typology of information on the financial market.

Based on its nature, information on financial market can be, in our opinion, of three types: (i) formal information; (ii) implicit information; (ii) bound information.

▪Formal information (FI) is about information available equally to all individuals, whether they operate or not on the financial market, namely that information available in the normative framework of society. This information is integrated ex ante into any decision/behaviour, in its entirety, immediately, and without search or risk costs (Nota bene: rigorously, only the formal information verifies the predicates of Fama’s available information in EMH).▪Implicit information (II) is about information that refers to events on the financial market, observable for all attentive, reflective, and interested economic agents (these qualities of economic agents do not imply any connotation on their financial competence) or what in the literature is called event information; implicit information is quasi-free, unlike formal information, because it involves, however, a cost of "transformation" (more precisely, of "translation") of the events observed into the (implicit) information usable in one’s own decision and behaviour.▪Bound information (BI) is about information that can (and must) be bought—either by ordering studies to companies specializing in financial market research, or, in illegal cases, by acquiring it through corruption, bribery or theft. Obviously, the bound information is not free, it involves both a search cost (e.g., payment of information acquisition studies) and the cost (coverage) of risk of discovering the illegal way of obtaining internal information from organizations.

Of the three categories of information, implicit information refers to behaviour. In other words, the implicit information is not actual information (like formal information or bound one), but an information deduced, inferred from observed behaviour of/on the financial market. This is exactly the meaning in which we say that, on the financial market, there may be a primacy of behaviour over information. The implicit information is therefore that information that the (potential) user of it produces him/herself, through idiosyncratic means: intuition (flair), rational calculation (e.g., CAPM, BSM, etc.), coincidence (luck, chance). 

Since the bound information is rather accidental (it is a constraint for agent not only from cost-benefit calculus perspective, but also from an ethical one), in what follows, we shall consider the formal information, and the implicit one only. As shown, the formal information (which is “pure”, that is, actual information) is, in fact, a quasi-constant information, because it is originated in the normative framework of the society—we prefer to call such a kind of information as geodesic information (in a similar sense of attractive force that is associated with the cosmological geodesic in the general theory of relativity). Despite its remarkable power to shape the behaviour, this power is relatively homogenous in time and space and, moreover, its influence is almost homogenous over all potential economic agents on the financial market. What makes the difference is, instead, the implicit information. Capacity to decipher the information from other agents’ behaviour—i.e., decisions, acts or abstentions, positioning and so on—is very different from an agent to another for obvious reasons. Consequently, the signification extracted from observations is necessarily different for different agents—as quantity, quality, signification, sense, speed of acquiring, etc. Since this difference is also held by the homo œconomicus model inside the neoclassic economic theory, we shall present below some crucial differences (Table 1):

Table 1 principles says that: (a) in the real world (real financial markets) the economic agents do not process the information as information, both because their bounded rationality and their lack of models of rationality, but that information which is revealed by intermediation of the observed behaviours; (b) it is not needed (just as it is not possible at all) to rationally process the implicit information from the observed behaviour, so any way, depending on the economic agents idiosyncrasies, it is (can be) used in order to capture, from the observed behaviours, the inputs are designed for their own behaviour; (c) the most important signification of the Table 1 is the following: the behaviour on the financial market is a second best one (or, as we prefer to say, a reachable best), which means that the behaviour is associated to the (economic, financial, monetary and so on) surviving purpose. This open door towards the evolutionary pattern of economic behaviour on the financial markets is in line both with the recent proposals (as the Adaptive Market Hypothesis—[11]) and with older quasi-evolutionary patterns provided by the behaviourism, or Behavioural Economics [12,13].

The adjustment of full rationality in the homo œconomicus model was based on recognizing the limits to such rationality (see Simon’s concept of bounded rationality), but which still remains in the informational territory, and, in addition, it is applicable to the same extent to all agents, while our proposal accepts differences between agents in being attentive, reflective, and interested in relation with the (observable) behaviour events on the financial market. Some other financial theories (as, for example, Efficient Market Hypothesis) also accept differences among agents, but such differences are linked to their competence to handle the pure information—consequently, there are sophisticated agents, who can process all available information, and non-sophisticated ones, who cannot make this, the last being so-called noisy traders. Our proposal, instead, “establishes” differences among agents not by putting them into two classes, as happened within EMH, but by a random distribution among agents of the capacity of being attentive, reflective, and interested. Figure 1 provides a synoptic view on the informational mechanism working on the financial market. In fact, the “translating” of the implicit information provided by the observable behaviours on the financial markets is possible because the observers see that the behavioural entropy is not yet (completely) established on those financial markets—such an establishment would mean no behaviour is performed or occurs. However, we do not agree, in this point, with some positions in the literature [14] according to which the entropy in economic process/system should be associated with the uncertainty—uncertainty is simply a ”black hole” from the perspective of observable behaviours or, the same, of implicit information on the financial market.

Nota bene: it is to be noticed that we have not here a simple herd behaviour—this last behaviour needs only attentivity, but neither reflectivity nor interest, for involved agents, that is, in Figure 1, only stages 1 and 2 are working in getting the own behaviour as a herd effect. 

### 4.2. On the Concept of Behavioural Efficiency of Financial Market

Based on the above allegations, we shall now introduce the concept of behavioural efficiency (BEF) of the financial market or, the same, of behaviourally efficient market hypothesis (BEMH). As it is well-known, the efficiency of the financial market is understood simply as the absolute impossibility to gain over the market average from transactions operated. This is an informational view, that is, it says that no available information can modify the chances to gain from transactions over the average, because, at any moment, all such information is already captured into the price (of course, we talk here about the capturing work done by the rational, i.e., sophisticated agents). In other words, market efficiency is viewed as the maximum homogeneity of information dissipation among agents at any moment. As it is known, the maximum of homogeneity is semantically equivalent to the maximum of informational entropy. Consequently, the efficient market hypothesis, for example, claims that, at any moment, the (financial) market stays at its point of maximum informational entropy. However, as we have already shown, such informational homogeneity could be accepted only regarding the formal information, based on the (realistic) assumption of a roughly equal attentivity capacity of agents regarding the market. As regarding the (different) reflective capacity of agents, or of their interest, it leads to different “productivities” in producing implicit information and, as a result, to different chances to exhibit a behaviour that can gain over the market average. There seems to arise here a new issue, namely that of behavioural efficiency of the financial market (BEF) about which some basic considerations are provided below: First of all, what does BEF mean? We think BEF is that state of the financial market when all significant agents have roughly the same attentivity and reflectivity capacity to extract implicit information from actual behaviours observed on the market. Such a definition is obviously only an adaptation, almost tale quale of the informational efficiency of the market. Consequently, some adjustments will be done further: (i) while the informational efficiency of the market is considered perfectly possible, the behavioural one is simply impossible (Nota bene: we remind that Grossman and Stiglitz showed that even informational efficiency is, in turn, impossible under the sanction of the disappearing of the financial market itself [15]); this impossibility of behavioural efficiency is grounded on the obvious fact that psychologically, intellectually, experientially, and culturally, the agents are irremediable different among them, and they cannot be reducible to a representative (i.e., medium) agent—as result, in a non-contextual way, the production of implicit information will be always different among agents and, so, the behaviour of them will always differ from one to other; the question of the representative agent still remains polemical here; (ii) the absolute impossibility of behaviour homogenization on the financial market, provided by the impossibility of equalization of the production of implicit information, means that the BEF is, in turn, impossible; (iii) the impossibility of BEF is not based on cost-benefit analysis (that is, on rational criteria), as the so-called Grossman–Stiglitz paradox of the EMH claims (from such a perspective, the G–S criticizing remains in the neo-classical economic territory, although, in our opinion, the paradox has, like our position, an absolute character, not a relative one), but it is originated into the realistic human condition of agents who operate on the financial market. There are, however, alternative ways to overpass the rigidity of EMH, either by an evolutionary adjustment [16], or by a sui generis combination between EMH and the behaviourism of Kahneman or Thaler type.Secondly, the BEF should be defined as that benchmark of the financial market at which no different distinct behaviour is possible, other than the already exhibited ones, at any moment. Is such a (asymptotic) tendency possible? This time we must claim the old herd behaviour and, in its slipstream, to introduce the concept of specific lazy riders. In fact, in the financial market, there always exists agents who are too lazy to be sufficiently attentive (and, much less, sufficiently reflective or interested) to extract implicit information from observed behaviours and who prefer to imitate the adopted behaviour by the agents who obtained sufficiently implicit information. Can such a phenomenon increase the behavioural homogeneity of the financial market? We would negatively answer this question. Different lazy riders will adopt different observed behaviours but, as the observed behaviours never come into their coincidence/homogenization, it results that the lazy riders approximatively keep the initial distribution of those behaviours on the financial market, because they will randomly adopt (and change) their preferred behaviours.Thirdly, the question can be posed if there are still agents who introduce noise on the financial market, and what the impact of such a noise is or could be. In standard financial theory (i.e., in EMH), the noisy traders provide exactly the informational niches which are (or can be) exploited by the rational/sophisticated traders. In our opinion, on financial market there are not, in fact, sophisticated vs. non-sophisticated agents/traders, but only agents with different capacity (either potential or actual) of attentivity, reflectivity, and interest towards the exhibited behaviours. Therefore, the lazy riders’ behaviour adds nothing to the potential implicit information to be extracted by the diligent agents, because the potential implicit information of their behaviour was already extracted from the behaviours which are imitated by those lazy riders.Fourthly, although the lazy riders do not create potential for new implicit information, could they, just by augmenting the number of a given type of observed behaviours, increase the probability (so to say) that the attentive and reflective agents extract the implicit information contained in that type of behaviour? Our opinion is negative: for an attentive and reflective agent, even only an occurrence of a behaviour type is sufficient to get the implicit information involved in it. Consequently, the massive occurrence of a given behaviour does not differ from a singular occurrence of that behaviour, from the perspective of the probability to extract the implicit information. Nota bene: perhaps, just by the contrary: the less often a behaviour is illustrated in practice, the more productive the implicit information it contains can be (probably we would talk here about a behavioural niche, analogously with the informational niche)—but this course of discussion will not be (for the moment) followed further.Fifthly, it seems to work on the financial market a kind of auto-feeding (technically: a positive feed-back) process of implicit information production: a behaviour leads to implicit information, which grounds a behaviour which, in turn, is observed and generates new implicit information and so on. Such a process necessarily must work in an asymptotically cushioned way. However, the state of affairs is not at all as such, because the implicit information extracted by an agent from an observed behaviour is not (qualitatively) the same implicit information which has grounded that observed behaviour—any agent has her/his idiosyncrasy, so the extracted implicit information is filtered by this idiosyncrasy and rather generates some “mutations” in the behaviour which will be shaped based on implicit information just acquired. This inaccuracy of passing the implicit information from a bearing-behaviour to another stays as the ground of the evolutionary model which must be (and which we shall) put of the entropy-based behavioural efficiency of the financial market—our main purpose of the paper. Nota bene: it would be wrong to make an analogy with the transcription or translating errors in Biology, so, we cannot speak here about hermeneutical errors, but, at most, about the inevitable filtering and altering of implicit information provided by the observed behaviours, which generate mutations in the future behaviour which, further, will be the object of the financial market selecting process (regarding this point of discussion, our position is approaching to Lo’s one regarding his conjecture called Adaptive Market Hypothesis, according to which the market selects the behaviours; also, our position is quite similar to that of Nelson and Winter regarding the concept of routine, at the organization level, that is also selected by the microeconomic market) [17].

Figure 2 tries to provide a suggestive synoptic map of the behavioural efficiency of the financial market, as discussed above.

## 5. The Proposal

Based on the concept of behavioural efficiency of the financial market, we come now to the heart of our paper—the concept of entropy-based behavioural efficiency of the financial market (EBBE). To this end, some preliminary discussion is useful.

### 5.1. On the Concept of Entropy-Based Behavioural Efficiency

#### 5.1.1. The Behavioural Entropy of the Financial Market

What could be something like behavioural entropy? The first thought refers, of course, to the concept of order. Behavioural entropy should address some degree of order of an entity (system/process/phenomenon/event), or, more applicably, some degree of homogeneity of that entity, from a pre-selected criterion. This first thought is not too different from the idea of informational entropy—namely, the degree of the possibility to use available information to gain over the market average. What is the state of affairs when the discussion is not anymore about information, but about behaviour? We shall try to point out our main opinions in this matter:Information can be homogenized to any degree, especially if on the financial market there are only sophisticated agents (that is, only they who count in integrating available information) and all of them have the same rationality potential or model, as standard financial theory claims;Instead, behaviour cannot be homogenized to any degree, because: (a) individual idiosyncrasies are irreducible; (b) some (probably, many) new behaviours which are based on the implicit information extracted from currently observed/old behaviours are altered in relation with their origin, so necessarily generate the increase of heterogeneity of behaviours. This idea—an automatic reversal process that opposes by itself to the indefinite increase of the behavioural homogeneity, that is, of behavioural entropy—deserves some extra comments:(i)Generally (especially after Prigogine introduced the concept of dissipative systems related to the entropy) it is accepted that, in the open systems (such as financial markets, for example) the inexorability of entropy increasing is, at least partially, off-set by the dissipative properties of those open systems—meaning that they can reduce or, at least, maintain the (low) level of entropy by throwing (more) high entropy in their environment (here, the model of Maxwell’s demon is very illustrative).(ii)However, it seems to us there is here an endogenous mechanism that can slow down (if, at limit, cannot reduce) the behavioural entropy or, more exactly, the behavioural-based entropy.(iii)We can now provide a more precise signification of the behavioural entropy: behavioural entropy is the ”reserve” (stock) of behaviours that can still be inferred as useable, by intermediation of the implicit information, from the currently observed behaviours—the larger that ”reserve”, the lower the behavioural entropy.(iv)From such a ”definition” of the behavioural entropy, we can extract the following idea: principally, the behavioural entropy cannot be objectively (that is, inter-personally) measured, as for example, the informational entropy is. In fact, the behavioural entropy level is inferred, by each economic agent (participant) in the financial market transactions, and the inference itself is proven just by performing a transaction (or by adopting a trading strategy, after the case).In fact, unlike the phenomenological entropy (in Thermodynamics), where there is an absolute time arrow—the permanent and spontaneous increase of the entropy in a closed system—and (partially) unlike the statistical or informational entropy, where the “evolution” of the probability distribution of states moves the system towards a higher level of entropy, the dynamics of behavioural entropy is ambiguous—it can either increase or decrease in relation to a fixed benchmark. Of course, it is of the highest interest to examine if there is something such as fixed points in the behavioural entropy trajectory which could be, from analytical perspective, either an *attractor* or a *source*. Such a direction of research seems to be very productive, taking into account that, unlike information, behaviour can be easily described as a trajectory. However, we do not agree with the idea that the so-called Complexity Economics or, worse, Econophysics or Quantum Economics, constitutes breakings with the economic theory mainstream—in fact, all three approaches remain inside the neoclassical economic theory. For example, the predicate of complexity, that is broadly understood as signifying to some a degree of difficulty in describing the causal relationships constitutes (Nota bene, to be noticed that such a difficulty has historical and instrumental conditionalities, it is not absolute), in our opinion, simply the unpredictability of a process, event, or system. What can introduce the unpredictability in the economic/social systems like the financial markets? One single thing: the free will of the economic agents. Is it to be observed that the free will goes beyond rationality or outside rationality—while the informational entropy is irremediably captured by the rationality—so the concept of behavioural (or behavioural-based) entropy is a more realistic one, because the behavioural entropy is generated only by (actual) behaviours? Therefore, a system can be very complicated (not complex) but remains simplex if the free will is not present within, and, vice-versa, a system can be very simple (not complicated at all) but it is complex if the free will is present within [18].Therefore, the behavioural entropy expresses a measure of the “concentration” of behaviours—the higher this concentration, the lower the behavioural entropy and vice versa. Obviously, the behavioural entropy replaces the more well-known informational entropy, by introducing the following changes (both conceptual and methodological): (a) what counts (as a novelty, besides the formal information) for the shaping of the own behaviour of an agent is, in fact, the implicit information, but the implicit information only appears to that agent as embedded into the observed behaviours on the financial market, so what is crucial here is the behaviour, not the pure information; (b) it is (very) possible that other agents on the market do not use all implicit held or acquired information, for different reasons (which, in turn, are, of course, unobservable for others) and, consequently, the implicit information on the whole level of the financial market could record a loss. From an evolutionary perspective, we have here a phenomenon of micro-selection—operated on implicit information, either consciously or unconsciously, by the agents themselves—which is subsequently concatenated to a macro-selection—operated by the financial market, in an impersonal way, on exhibited (actual) behaviours. Therefore, from the point of view of behavioural entropy, on the financial market work together both a micro-selection (on gained implicit information) and a macro-selection (on exhibited behaviours). Nota bene: it is not logically needed that the micro-selection be done through the rational tool, that is, based on models of rationality such as within EMH, for example; (c) so, the behavioural entropy is a result of two interwoven factors, in fact two distinct invisible hands: (1) the micro invisible hand—mIH (namely, the micro-selection), and (2) the macro invisible hand—MIH (namely, the macro-selection).

The idea of the two interwoven factors—mIH, and MIH, respectively—in generating the behavioural entropy (and, further, the behavioural efficiency of the financial market) is, in our opinion, relevant for at least two reasons:(i)Therefore, provided ”procedure” or a mechanism by which, actually, the implicit information is transformed/translated/converted into own behaviours by the attentive, reflective, and interested economic agents is provided. This issue is very important, because, as it is well-known, EMH (and, partially, AMH) are criticized exactly because they do not deliver a mechanism by which their (informational) efficiency is actualized. Such a procedure, combined with the structure of information proposed for the financial market (another upbraiding to the two models of financial market) could make a step further in modelling the (real) financial markets.(ii)However, the most important consequence of the mIH-MIH mechanism—Nota bene: we could abbreviate this mechanism as (mM)IH—consists of supplying a channel for empirically/factually testing the behavioural-based entropy hypothesis, in combination with the behavioural efficiency of the financial market. A synoptical image of this potential of the mechanism (mM)IH is presented in Figure 3.

#### 5.1.2. Briefly, on the Naked Phenomenology of the Behavioural Entropy

First of all, it must be said that regarding the behavioural entropy we cannot speak other than inside the social field (with its species: economic or financial markets fields). The reason is quite simple: all economic/financial/social events are effects of the human being behaviour—such events occur, develop, and disappear if and only if an individual or a group of individuals perform a behaviour. Why the implicit information, that is, the behaviour, has a primacy over information in the financial markets, has been already argued above. 

Secondly, the question that can be put regarding the added value of such a concept—behavioural entropy—related to the currently generally accepted concept of informational entropy is: why is it necessary to pass from informational entropy to the behavioural one? We think the “citizenship” of the concept of behavioural entropy is justified at least by the following reasons:In fact, no informational entropy can be empirically tested if the economic agents involved do not perform at least an action (either act or abstention). Consequently, the integration of all available information on the financial markets (as, for example, EMH claims) cannot be observed at all. Therefore, the reason here is the observability of the entropy on the financial markets—the only observable entropy is provided by the behaviour, that is, by the behavioural entropy.The concept of behavioural entropy seems to solve also the Grossman-Stiglitz paradox regarding the informational efficiency in the financial markets, inside the EMH model. Indeed, according to the above mentioned quasi-automatic reversal of the behavioural entropy tendency to increase and, also, according to the (mM)IH mechanism, it seems that the financial markets never reach their maximum of behavioural efficiency (and, correspondingly, their maximum behavioural entropy).Briefly, the behavioural entropy means the degree in which new behaviours are available and practicable, as this possibility spectrum is provided by the implicit information obtained by the attentive, reflective, and interested economic agents (see Figure 3). The behavioural entropy signifies the degree of behavioural heterogenization of the financial markets. To be mentioned here is the position held by Lo: the financial market selects the trading strategies (or, often, the individual transactions performed within a given trading strategies), namely, what we understand by the MIH part of the (mM)IH mechanism (Figure 3).

Synoptically, Figure 4 shows (rather as a qualitative intuition, but which could be analytically developed, with the sufficient and necessary arguments in case) the essence of the concept of behavioural entropy as we would want to introduce here (where: IEN means informational entropy, IEF means informational efficiency, BEN means behavioural entropy, BEF means behavioural efficiency).

#### 5.1.3. Combining the Behavioural Entropy with the Behavioural Efficiency

By combining the concept of behavioural efficiency (the relative scarcity of distinct accessible behaviours) with the concept of behavioural entropy (the relative concentration of distinct exhibited behaviours) we can obtain the desired objective of the paper: the concept of the entropy-based behavioural efficiency of the financial market (EBBE). We shall design this concept as follows:The qualitative relationship between behavioural efficiency (BEF) and behavioural entropy (BEN) is directly proportional: the higher behavioural entropy, the higher behavioural efficiency, based on the following reasoning: (i) high BEF means many behaviours available (presupposed to be, also, accessible), so the financial market in this case has a high degree of homogenization; (ii) a high degree of homogenization means, in turn, a small degree of concentration, that is, a high BEN (of course, the reciprocal reasoning is true too); the same for the case in which on the financial market there is a low BEF.The quantitative relationship between BEF and BEN is (or should be), in our opinion, directly non-linearly proportional—more precisely, it is a logistic curve, where BEN is the independent variable, in order to exactly get the searched EBBE (Figure 5 graphically expresses such a conjecture).

Nota bene: The continual vs. discrete nature of the EBBE curve is, of course, a matter of a methodological convention, in relation to the type of the involved mathematics.

Introducing the ontological impossibility areas is (macro) qualitatively justifiable as follows (Nota bene: to be mentioned is that the specialty literature makes, usually, conceptual distinctions between ontological entropy that is understood in Georgescu-Roegen’s sense—that is, as an energetic flow from the Sun’s energetic fund—and the metaphorical one, which we agree, together with others researchers, to be associated to some order of interest for the economic/financial processes [18,19]):‒The bottom area means there is not a market—that is, it refers to a quasi-autarchic state of affairs.‒The upper area means too much specialization of agents, which, in fact, leads to a too high fragmentation of the market. Nota bene: the market must have some granulation/agglutination to actually work. The very idea of the necessary granulation of the financial market is linked both to the financial market structure (e.g., the degree of oligopoly) and to the normative framework of the society as a whole. Additionally, the (mM)IH mechanism proposed above tries to capture this idea into the benefit of the concept of behavioural-based entropy.‒The left area means a stiffening of heterogenous behaviours structure which simply blocks any tendency to increase the homogeneity.‒The right area means a stiffening of the homogenous behaviours structure which blocks any tendency to increase the heterogeneity.‒Although Figure 5 seems going, to some an extent, beyond the (specific) ”rationality” of the financial market, in fact, it prepares the background for the below reasoning about the automatic stabilizer regarding the relationship between behavioural entropy and the behavioural efficiency in the financial markets.Why a logistic curve? We shall present below the main reasons for which the expected entropy-based behavioural efficiency happening on the financial market is (and must be) logistically shaped:(a)When the BEN is small (that is, it is near the point below which it cannot anymore, ontologically, decrease) and it just begins to increase, this means that the degree of concentration of behaviour types decreases correspondingly; however, the process of deciphering the new structure of behaviours takes some time—or, equivalently, generates some lags—so the BEF starts to slowly increase its slope from an initial null one, that is, from the bottom asymptote.(b)As BEN continues to increase, more new behaviours (generated by continual decrease of the degree of concentrations of behaviour types) are observed and additionally implicit information are captured which grounds new distinct behaviours—therefore, an auto-catalytic phenomenon appears in the BEF “territory”, which supplementarily augments the dynamics of BEN and so on; technically, the marginal implicit information provided by the observed behaviour types is increasing.(c)Based on the above considerations, EBBE curve has, in its first part, a convexly increasing shape, which means that BEF is acceleratingly increasing.(d)At a given point of the BEN (technically, the inflexion point of EBBE curve), the convexity is replaced by the concavity of the EBBE curve, which means that, from this point (Nota bene: as known, on the vertical axis, this point represents the real solution of the secondary derivative of EBBE function) to right, the BEF will deceleratingly increase until to the upper asymptote; this change in convexity is justifiable as follows: beyond the inflexion point are already enough distinct behaviour types on the financial market, so the marginal implicit information extracted from the observed behaviours is decreasing.(e)As conjectured above, the maximum of BEN gives the maximum of BEF. This is not different, however, from what is happening inside EMH assumptions: the maximum of information homogenization, that is, the maximum of informational entropy, leads to the maximum of market efficiency.

#### 5.1.4. On a Possible Invariant in Behavioural Entropy Dynamics

Unlike other species of entropy, the behavioural entropy has not a time arrow, that is, it does not irreversibly come towards the upper asymptote in Figure 5. In fact, we believe there is a no man’s land, shaped like a rectangular band around inflexion point (both vertically and horizontally, not necessarily as a quadrat) which we could name as the osmotic behavioural entropy area of the EBBE curve. As we shall describe below, the osmotic behavioural entropy area (OBEA) works as a paired automatic behaviour stabilizer (PABS)—the two components of that pair are, of course, BEN and BEF. (Figure 6 provides synoptic help). The idea of a concomitant existence of entropic and neg (or anti)-entropic ”forces” which act (including in the economic/financial field) to lead the economic/financial dynamic of markets is not completely missing in speciality research, although it is not developed as a mechanism like our OBEA [16].

### 5.2. A Simple Generic Formalization

The mechanism described in Figure 6 regarding the PABS working can be formalized in a simple way in order to help the formulation of our proposal (we shall ignore the ontological impossibilities, without compromising the reasoning carried out—Nota bene: one can at any point reintroduce the impossibility areas into formalism simply by appropriately introducing a constant, vertically and/or horizontally, according to Figure 6), taking into account the following initial constraints: $BEN\in \;\left(0,1\right)$, and $BEF\in \;\left(0,1\right)$.

First of all, by noting with $U\in \;\left(0,1\right]$ the upper asymptote value, and simplifying notations for BEN:=x, and for BEF:=y, then the logistic equation of the EBBE curve is written as: (1)$$y=\frac{U}{a+b\cdot {𝓯}^{-k\cdot x}}$$
where 𝓯 is the so-called Feigenbaum ratio/limit/constant, given by the ratio of two consecutive distances where bifurcations happen in a chaotic non-linear process; its numerical value is 4.67 and expresses, in essence, a similarity (scalable) factor in the chaotic processes; a, b, k are calibration factors aimed to fit the numerical values into qualitative assumptions; for the cases in which it must be calculated $y^{-1}=x$, we have:(2)$$x=\frac{1}{k}\cdot \log_{𝓯}\frac{b\cdot y}{U-a\cdot y}$$

## 6. Discussion

The following seven issues are chosen to be discussed below in order to better clarify our proposal: (1) why, after all, a logistic curve is (the most) appropriate to formally capture the relationship between behavioural entropy and behavioural efficiency on the financial market; (2) which are the Cartesian coordinates of points M, N, and P in Figure 6; (3) how to measure, in fact, the behavioural entropy based on financial market functioning, that is, how to transform the behaviours observed into implicit information; (4) how to measure the behavioural efficiency as an endogenous variable of the EBBE function; (5) questions regarding the (factual) testability of our hypothesis/conjecture; (6) how the jumps both in BEN and BEF work; and (7) why the Feigenbaum ratio/constant as the base of exponential (or, equally, of the associated logarithm) for EBBE function is used.

(1) There is a general reason for which, in the social field, but especially where the behaviour is involved, the logistic curve is not only the most appropriate way to model the processes/phenomena, but it seems there are not credible alternatives to it. Indeed, when the behaviour is involved, a time interval to understand an emerging state is needed, so the reaction to that emerging is lagged or slowed down—in fact, a kind of inertia manifests here. Then, as the agent familiarizes with the new environment, the speed of reaction increases, so the EBBE curve is convexly shaped in its initial stages. Our concept of implicit information, based on which the paper was built, comes to additionally improve this explanation: the more behaviours emerged on the financial market (that is, the higher behavioural entropy is installed), the more implicit information is available and, proportionally, more of them is deciphered and used in designing new own behaviours—so, an acceleration of behavioural efficiency takes place; in this phase, the marginal implicit information is increasing. However, after a while, namely after the inflexion point, so many behaviours are already working on the financial market that the difficulty to observe and interpret those behaviours becomes significant and the marginal implicit information gained is decreasing (or, equivalently, the EBBE curve becomes concave).

(2) The coordinates of points M, N, and P in Figure 6 are of the most importance and relevance in our proposal because they not only delimitate the osmotic behavioural entropy area (OBEA), but also underpin the designing of the empirical tests of the proposal. We shall sketch here some considerations regarding this issue:Regarding point P: essentially, the abscissa of P signifies the minimum of possible behavioural entropy on the financial market (Nota bene: on the financial market, the behavioural entropy is never null—be notified that the null value accepted below is only conventional, for algebraic reasons). Moreover, that abscissa signifies the start of OBEA, that is, the triggering of the paired automatic behavioural stabilizer (PABS) of behavioural entropy (and, through EBBE function, of behavioural efficiency, too). Correspondingly, the ordinate of P means the minimum of possible behavioural efficiency on financial market—that is, analogously to the informational efficiency of EMH, the maximum of chances to beat the market by the exhibited behaviour. Additionally, this point starts OBEA from the efficiency perspective.Regarding the point N: the abscissa means the maximum of possible behavioural entropy on the financial market, that is, the point where the behavioural entropy jumps to its initial point (namely, point P) for reasons which, we hope, will become clearer below. The ordinate of point N means, as the one of point P, the minimum level of behavioural efficiency on the financial market. Therefore, why and how is jump 2 (see Figure 6) possible and what signification does it bear? Such a jump, from the maximum level of behavioural entropy to its minimum level is possible in a single case: the arising of a new financial paradigm which recommends or allows a new type of behaviour on the financial market. Such an event suddenly and almost completely excludes the old behaviours (especially from their quantitative perspective) and installs a new paradigm, namely a new trading strategy (Nota bene: it is not necessary, of course, for a single new behaviour to be established, but even a few). Here, obviously, is large room to develop an analysis on the concept of the Kuhnian paradigm [20] applied to financial theory—especially from the perspective of hiding the anomalies under the mat, or regarding the incommensurability between successive financial paradigms.Regarding the point M: the abscissa of point M signifies the maximum value of the behavioural entropy, and its ordinate also signifies a maximum, namely that of the level of behavioural efficiency. The point M has the role of triggering the functioning of the paired automatic behavioural stabilizer (PABS)—as shown, this device acts in two steps: (a) jump 1, behavioural efficiency comes down to its minimum level; (b) jump 2, behavioural entropy comes to the left of its minimum level (in fact, PABS resets the EBBE mechanism for a new cycle of behaviours on the financial market).

(3) Behavioural entropy is obviously different from other concepts of entropy which are also assigned to the social field. Generally, the society can be evaluated from the entropic perspective trying to identify (and to measure, if possible) the degree of normative order within which society is functioning. In the case here discussed, the order/disorder of the financial market should be associated with the degree of homogeneity regarding the behaviours exhibited in relation to trade acts/abstentions or, more generally, in relation to the trading strategies implemented. As shown in Figure 6, the behavioural entropy can either increase or decrease—the increasing shape is logistic, while the decreasing one is rigid against abscissa. When new distinct behaviours emerge based on the implicit information deciphered by agents, the space of financial behaviours becomes more homogenous, so we must interpret this process as one which implies an increase of the behavioural entropy, and as the opposite when current financial behaviours disappear. Let us note: $B_{i}$ is the number of distinct behaviours (for example trading strategies) that are currently working on the financial market at moment i, $B_{i}^{+}$ is the number of new distinct behaviours emerged on the financial market at moment i, $\alpha_{1}$ is the rate of converting/translating observed behaviours into implicit information, $\alpha_{2}$ is the rate of emerging new distinct behaviours based on the implicit information acquired ($0\leq \alpha_{1}\leq 1$, $0\leq \alpha_{2}\leq 1$), $\delta_{N}$ is the abscissa of point N, and $\delta_{i}$ is the abscissa of $B_{i}$. Then, we can write:(3)$$\alpha_{1}=\frac{II_{i}}{B_{i-1}}$$
(4)$$\alpha_{2}=\frac{B_{i}^{+}}{II_{i}}$$
(5)$$B_{i}^{+}=\alpha_{1}\cdot \alpha_{2}\cdot B_{i-1}\cdot \left(\frac{\delta_{N}-\delta_{i}}{\delta_{N}}\right)=k\cdot B_{i-1}$$
where $k=\alpha_{1}\cdot \alpha_{2}$ (that is, it signifies the rate of birth of behaviours), and $\delta =\frac{\delta_{N}-\delta_{i}}{\delta_{N}}$. As Figure 5 shows, when $\delta_{N}=\delta_{i}$, the behavioural entropy jumps back to $\delta_{P}$, and the cycle (0)–(1)–(2) in Figure 5 restarts. 

Of course, in a time period there are both births and deaths of behaviours. The disappearing of behaviours is caused by their inefficacy selected as such by the financial market itself [20,21]. If noted with $B_{i}^{-}$ the number of behaviours disappeared at moment i, then:(6)$$B_{i}^{-}=\beta \cdot B_{i-1}\cdot \delta$$
where β is the rate of death at the moment i:(7)$$\beta =\frac{B_{i}^{-}}{B_{i-1}}$$

As we see, the function of the disappearing of behaviours also holds the variable δ—indeed, the death of behaviours is directly proportional with the “density” of the behavioural space in the financial market, because of the (frequential) probability that a given behaviour will be rejected by the “supreme” selector—that is, by the financial market—increases when that density increases. As a result, we can write:(8)$$B_{i}=B_{i-1}+B_{i}^{+}-B_{i}^{-},\;\mathrm{or},\;\mathrm{equivalently},$$
(9)$$B_{i}=B_{i-1}\left[1+\delta \left(k-\beta \right)\right]$$

Based on the above considerations, we propose to understand by the behavioural entropy the ratio $BEN_{i}=\frac{\delta_{N}-\delta_{i}}{\delta_{N}-\delta_{P}}=\frac{\delta \cdot \delta_{N}}{\Delta }$, where $\Delta =\delta_{N}-\delta_{P}$, that is, a normalization to unity of the possible values of the number of working behaviours at the moment i. As presumed above, the behavioural entropy moves inside the interval $\left(0,1\right)$. If noted $\lambda_{i}=\frac{B_{i}}{B_{i-1}}$, then we have:(10)$$\lambda_{i}-1=\delta \cdot \gamma$$
where γ=k−β is the net growth of the number of distinct behaviours (for example, trading strategies) on the financial market. Therefore,
(11)$$BEN_{i}=\frac{\lambda_{i}-1}{\gamma }\cdot \frac{\delta_{N}}{\Delta }$$

For Δ and γ fixed (constant), the behavioural entropy evolves linearly with the growth index of the number of behaviours on the financial market.

Nota bene: the temptation to adapt the Lotka–Volterra equations of the predator–prey model to the issue of behavioural entropy on the financial market is almost irresistible. In our opinion, it is not the case, however: the disappearing of the behaviours is not linked to the emergence of new behaviours (as the population of prey is dependent on the population of predators, and vice-versa), except, however, the fact that the probability of disappearing seems to be, to some extent, proportional with the emergence of new behaviours, but there is here only a general conditionality, not a genuine causality.

In fact, we have now to show how the behavioural efficiency is generated—through the logistic curve—and how can it be measured.

(4) We come now to the issue of behavioural efficiency on the financial market. Regarding informational efficiency is not the case to insist here, because this concept is well-known—essentially it says that no available information can be used to improve the agent’s success on the market, that is, to beat the market (more precisely to gain over the market average). It is quite obvious that the informational efficiency addresses the information as such, for example, prices, productivity and so on [11], while it does not consider the implicit information as, instead, our paper does. We shall shortly describe the behavioural efficiency as follows:Firstly, we need a benchmark against which it can be judged if there is or is not a behavioural efficiency on the financial market (for the informational efficiency, that benchmark is the market average price or gain). What can be said regarding the behavioural efficiency benchmark? Since our driving variable in the EBBE model is the implicit information, and the implicit information leads to new distinct behaviours, it seems that the looked-up benchmark could be the covariance between the distribution of price and the distribution of number of behaviours: if the covariance is positive, then the dynamics of number of (distinct) behaviours on the financial market directly follows the dynamics of the price. This means that the implicit information extracted from the behaviours actually observed is still significant and leads to designing and implementing new behaviours (new trading strategies), so a positive covariance signifies an increase of behavioural efficiency. Inversely can be the reasoning for the case when the covariance is negative, that is, this time we have a decrease of the behavioural efficiency. Nota bene: if p stays for the price, and b for the number of distinct behaviours on the financial market, then, for n observations, the covariance in this case is (as is well-known) written as:(12)$$cov\left(p,b\right)\;=\overset{n}{\underset{j=1}{\sum }}{𝓅}_{j}\cdot \left(p_{j}-E\left(P\right)\right)\cdot \left(b_{j}-E\left(B\right)\right)$$
where with ${𝓅}_{j}$ is noted the (frequential) probability and $E\left(X\right)$ is the mathematical expectation of the random variable X. If the probability is equal (that is ${𝓅}_{j}=1/n$, for any j), the covariance is equivalently writable as:(13)$$cov\left(p,b\right)\;=\frac{1}{n}\cdot \overset{n}{\underset{j=1}{\sum }}\left(p_{j}-E\left(P\right)\right)\cdot \left(b_{j}-E\left(B\right)\right)$$

Nota bene: in our opinion, on the financial market (as is the case generally in the social or economic field [17], where the events are unrepeatable/singular), the only possible probabilities which can productively work are the Bayesian ones; therefore, the equality of the probabilities (${𝓅}_{j}=1/n$) is required (and justified as well) by the Laplacian principle of insufficient reason, which, in turn, is generated by the (almost complete) incertitude (not indeterminacy!) on the future in the social or economic field. 

Secondly, although the tool of covariance can give us a presumption about the dynamics of the behavioural efficiency, in fact, (the level of) this efficiency is already given by the EBBE curve—it is sufficient to calculate the behavioural entropy and to introduce it into the EBBE function (of course, in the conditions of previous specification of the empirical values for the parameters of that logistic function).Thirdly, it is much more important to provide the signification of the dynamics of the behavioural efficiency (of course, inside the OBEA):‒When the behavioural efficiency increases, this means that the implicit information generates new (distinct) behaviours which compete with each other to gain the market rewards. Consequently, the chances for gaining those rewards are more largely shared and dissipated, so they become less attractive or more costly (regarding the effort to extract more implicit information) (Nota bene: the test for any agent regarding the decision to keep or abandon a behaviour—see the β coefficient above—remains, without doubt, the market reward).‒when the behavioural efficiency decreases, the mirrored phenomenon emerges and, as a result, the market rewards are more narrowly shared and dissipated, encouraging the agents to keep (but also, in some conditions, even to abandon when other better behaviours can be adopted) the behaviours successfully tested.

(5) Our hypothesis—that could be called the entropy-based behavioural efficiency hypothesis (EBBEH) or conjecture (here we follow the concepts of hypothesis/conjecture assumed by Popper [22]—should be empirically testable (that is, should be falsifiable), in order to verify its condition of scientificity. We shall provide some ways/tools which could support/help the scientists in the matter to test this hypothesis, by addressing the main empirical implications of the EBBE curve:

Regarding the parameter U (upper asymptote): the parameter U is placed in line with the bottom level of the upper vertical area of ontological impossibility of the behavioural efficiency. Consequently, it should be determined outside the EBBE ”territory”, based on specific reasons regarding the possibility/impossibility of the financial market to work if the behavioural efficiency increases over this limit. Such a specification is, therefore, not a theoretical issue, but an empirical one (perhaps, however, not too local [23]); here we have roughly the same question that was handled by Grossman and Stiglitz in relation with the EMH.Regarding the coordinates of points M, N, P: the question of the coordinates of the three points which delimitate the OBEA constitutes not only an empirical issue, but even a terrain of separate conjectures. In our opinion, this issue must be treated and solved before and independently from any testing of the EBBE hypothesis. For example, the degree of concentration in calculating the behavioural entropy cannot be smaller than $1/\sqrt{n}$, where n is the number of distinct classes of behaviours on the financial market, which gives some information about $\delta_{P}$.Regarding the parameters a, b, k: these parameters should be established based on pure mathematical conditionality—more precisely, they may ensure on the generic shape both of EBBE curve and of OBEA (see Figure 6). Regarding $B_{i}$: although this variable seems to be extremely difficult to estimate, this is not at all so. In fact, currently, there are many such ”lists” of trading strategies used in financial market transactions. Of course, rigorously establishing a set of criteria aimed to deliver the list of trading strategies (that is, behaviours) accessed at a moment (or within a time interval) could be further necessary, but in principle, this question is not, properly speaking, a problem.Regarding the testability/falsifiability itself: a possible “algorithm” to be used in empirically testing our hypothesis embedded into the EBBE model of the financial market could be as follows.‒Formulating the prediction on the behavioural efficiency:▪Establishing OBEA (independently from EBBE model);▪(Purely algebraically) calibrating the model for parameters a, b, k;▪Calculating $\lambda_{i}$;▪Calculating the degree of behavioural entropy—$BEN_{i}$;▪Formulating the prediction on the behavioural efficiency, that is:(14)$$BEF_{i}=y\left(BEN_{i}\right)$$
or sometimes:(15)$$BEF_{i}=y\left(BEN_{i-1}\right)$$‒Generating the time series needed for empirical behavioural efficiency;‒Calculating the empirical/factual behavioural efficiency;‒Comparing the predictive proposition with the descriptive one and deciding on the corroboration vs. refutation of the prediction.

(6) The two jumps phenomena are quite exotic in our proposal. Here, we shall try to provide some extra justifications aimed at increasing the credibility of such “ingredients” of the EBBE hypothesis:Regarding jump 1: when the point M of OBEA is reached (that is, when the behavioural entropy as well as the behavioural efficiency are both at their maximum), a new paradigm is about to be established (any case, its early signals are already perceptible), which means that the efficiency of the financial market is about to (possibly) catastrophically decrease until the point N. In fact, the spectrum of irregular and big rewards within the new paradigm (which, it must be noticed, at its initial road, contains very few trading strategies—possibly one only), leads agents to abandon in mass the old behaviours of financial transactions and try to access the new one/s, which is exactly the logical content of behavioural efficiency decreasing.Regarding jump 2: once the N point is reached, its abscissa becomes non-compatible with its corresponding ordinate, so the behavioural entropy must accommodate with the new level of behavioural efficiency—so, the content of the jump 2 is the moving of the EBBE curve form N point to P point.

Nota bene: although our description of the two jumps seems to suggest that jump 2 waits for jump 1 to produce itself, in fact, the two jumps happen almost simultaneously. Of course, jump 2 needs jump 1 to be already done, but it emerges in a very short time—the time interval needed for the majority of agents to become aware about the crash of the behavioural efficiency on the financial market. From the testability point of view, we think that this phenomenon of jumps is appropriate to be empirically tested, so constituting evidence (or a counter-evidence) of our proposal/hypothesis. The jumps, together with the PABS, seem to evoke the interesting (and quite stranger) concept and mechanism of the hypercycle.

(7) Generally, the exponential base in the logistic functions is e (the base of natural logarithms, namely e=2.718). From calibrating reasons, of course, any base could be adopted (verifying the definitional conditions of logarithms). By choosing the Feigenbaum ratio/constant we want to send a message that the issue of the entropy-based behavioural efficiency on the financial market could be approached, with great productivity, from the perspective of the Chaotic Theory (non-linear dynamical systems)—for example, by establishing/measuring some time interval in which equal rates of $B^{+}$, $B^{-}$ or other variables of the EBBE model happen. Moreover, the EBBE model could be used to verify both the universality of the Feigenbaum ratio or, if we are lucky enough, to discover another such ratio, perhaps specific for the financial market.

## 7. Results

The main results of the research presented in the paper are: (1)agents in the (real) financial market are behavioural-driven rather than informational-driven;(2)there are three sorts of information in the financial market: formal, implicit, and bound, and all behaviours are performed around and based on these types of information, especially on the implicit one;(3)the basic (and crucial) information which counts in the financial market functioning is the implicit information, and it is hermeneutically extracted by (attentive, reflective, and interested) agents from observed (actual) behaviours;(4)the financial market has entropy, which is measured based on the density of financial space regarding the number of distinct behaviours (trading strategies) which are observable, interpretable, and designable based on the implicit information they exhibit;(5)the entropy on the financial market is not an informational entropy, but rather a behavioural one—the behavioural entropy measures the degree in which the financial market shows its heterogeneity regarding new possible and available behaviours;(6)the behaviour on the financial market is entropically-driven [24], in the sense of behavioural entropy;(7)the financial market is behaviourally efficient rather than informationally efficient—the concept of efficiency, either as informational or behavioural, has the same signification: exhaustion of the (praxiological) occasions to perform behaviours;(8)the formal relationship between behavioural entropy and behavioural efficiency is (or can be conjectured as) logistic (taking into account the general behaviour of the economic homo œconomicus, which is very different—and much more realistic—from the mathematical homo œconomicus);(9)the base of the involved logarithm is (as proposed to be) the Feigenbaum ratio;(10)the signal of behavioural efficiency on the financial market is the covariance between price and the number of distinct classes of behaviours which are actually working (and are observable by intermediation of the implicit information);(11)the behavioural entropy (as exogenous variable), and the behavioural efficiency (as endogenous variable) moves only inside a bounded bi-dimensional area called the osmotic behavioural entropy area, which is functioning based on a paired automatic behavioural stabilizer, so the behavioural entropy as well as the behavioural efficiency do not have a time arrow (as the informational entropy has, instead);(12)in the EBBE (entropy-based behavioural efficiency) model of the financial market, there is sufficient conceptual room to accept an adjusted (at a range from cognitive field to praxiological one) Kuhnian paradigm, linked to the paired automatic behavioural stabilizer.

## 8. Conclusions 

The behavioural driver on the financial market seems to better explain the actual functioning of this market than does the informational driver, which still currently dominates the financial theory. In addition, the behavioural entropy and the behavioural efficiency in the financial market could be more easily (and, possibly, irrefutably) tested empirically (e.g., by the (mM)IH mechanism), of course based on Popperian recommendation: formulate the least likely conjectures regarding both the micro-selection and the macro-selection among the accessible behaviours, as they are suggested by the implicit information.

The EBBE model allows much more directly and productively the evolutionary approach of the financial market, as Andrew Lo and others do today. Since the implicit information, which drives the behaviour in the financial market, is always connected (at least) to the formal information, the EBBE model is, in fact, a model immersed into the society normative framework and, even more, in the society cultural geodesic, that is, in the society value matrix.

The behavioural entropy, together with its correlative, the behavioural efficiency in the financial market, could lead to the conclusion that the Thermodynamic entropy—so de-connected from the human behaviour—could be at most a remote benchmark for the (real) entropy in the economic/financial field. In fact, most of the dissipativity in society is of normative nature, therefore quite departed from the Thermodynamic entropy.

The complexity (in the sense of the theory of non-linear systems or, the same, of the Chaotic Theory) does not have any relationship with the complicatedness of the systems involved—simply, the complexity is ”delivered” by the presence of the unpredictability, while the unpredictability is caused by the presence of the free will. Therefore, the financial market (as any other social area) is complex no matter how non-complicated it is. In our proposal, implicitly, such an understanding of the complexity is taken into account by the (mM)HI mechanism, in which economic agents’ actions (that is, behaviours) ”play” inside the osmotic behavioural entropy area.

The Grossman–Stiglitz paradox, which is applicable to EMH (in fact, more generally, to any informationally-based financial theory) becomes superfluous in relation with the EBBE model of the financial market, because both the osmotic behavioural entropy area and the paired automatic behaviourally stabilizer ensure the self-regulation of the financial market from the behavioural efficiency perspective. In fact, the self-regulation of the EBBE model forbids, principally, to reach the maximum (complete) behavioural entropy as well as the maximum (complete) behavioural efficiency of the financial market. If a measure (most probably, relative) of the behavioural entropy could be found, then such a measure could be used as predictor [25] for the behavioural efficiency on the financial market—again, by opening a ”door” for empirically/factually testability.

Our proposal is, obviously, polemical and debatable. We ourselves shall continue to reflect on it, in order to improve—conceptually, logically, and methodologically—the present version. We think some directions of further research (or, at least, some subjects presumable to bring added value in the context of our approach) could stay in attention of the (interested) scientists in this matter:

The disappearing of behaviours (trading strategies) from the financial market could be, at least partially, a result of a sort of behavioural cannibalism—some behaviours are integrated in others, as a consequence of new relevant implicit information.In order to calculate the exogenous variable named behavioural entropy, a set of criteria to classify the trading strategies can be useful, together with a signal to warn about changes in this typology.Really, from 1965 (Samuelson’s martingale, as well as Fama’s EMH) were there jumps (inside our osmotic behavioural entropy area), and, if yes, what are their paradigmatical significations?Is there something of the Feigenbaum ratio on the financial marketing functioning, for example, linked to the time intervals in which the same changes of $\lambda_{i}$ are recorded? For example, some scientists claim a time interval between cause and effect and, on such a base, make a connection between entropy and causality, although we think that we have here simply a logical circularity, because by stating the time is a metric of causality is an… axiom.We think that our proposal can be, in its entirety, approached from the dissipative systems perspectives, taking into account that the concept of dissipativity (introduced by Ilya Prigogine) is founded on the concept of entropy as a measure of order.It seems to us that the Chaotic Theory has advantages compared to other alternative approaches, because it is possible that some fixed points (either attractors or sources) can be established about the behavioural entropy trajectory (or, equivalent, through the logistic function, about the behavioural efficiency trajectory).One of the most important issues (which was not discussed in the paper) is the formalism of translating the observed behaviour into implicit information, perhaps into different classes (if is the case) of such information. This direction of further research constitutes the first priority of authors in order to additionally consolidate the proposal made.A (perhaps utopian enough) direction to advance in the modelling of the financial market field is to develop a new mathematics (more general, a new quantitative formalization) capable of capturing the behaviour per se, not as a result of information processing inside rationality models. The authors think that the (algebraic) topology can be a good candidate in that matter—to be reminded that there is, already, a quasi-topological proposal to model the (inter)action [19], very similar to, although much more subtle logically, Feynman’s diagrams for the quantum inter-actions.Most likely, the Shannon informational entropy could be re-examined in order to prove that it is, rather, a behavioural entropy.

## Figures and Tables

**Figure 1 entropy-23-01396-f001:**
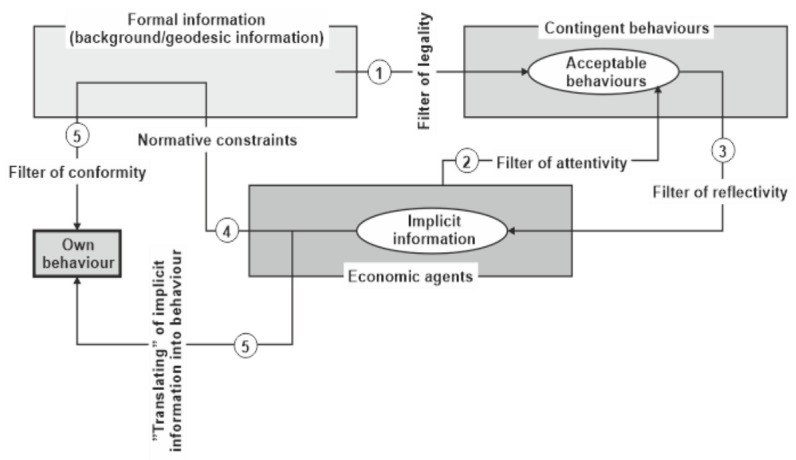
Getting implicit information from observed behaviour.

**Figure 2 entropy-23-01396-f002:**
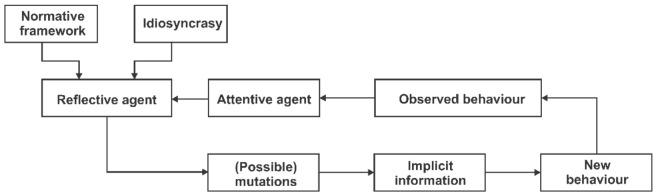
The circuit of behaviours (behavioural efficiency). Source: authors’ graphical construction.

**Figure 3 entropy-23-01396-f003:**
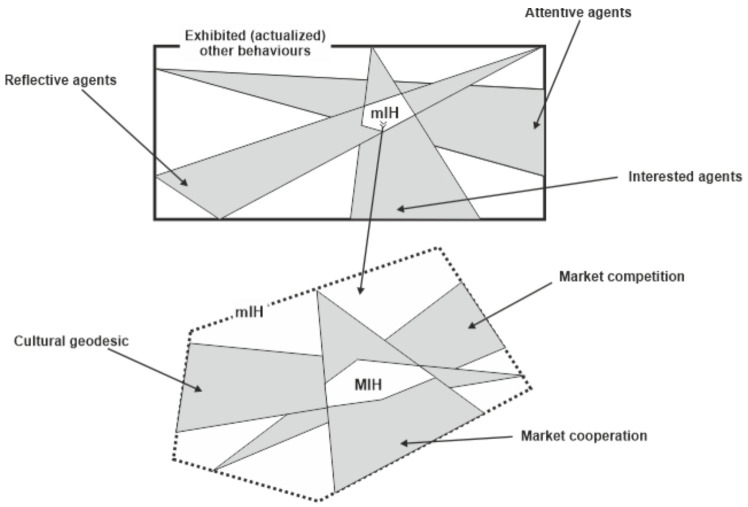
The mechanism (mM)IH by which the behavioural-based entropy can be empirically tested. Source: authors’ graphical construction.

**Figure 4 entropy-23-01396-f004:**
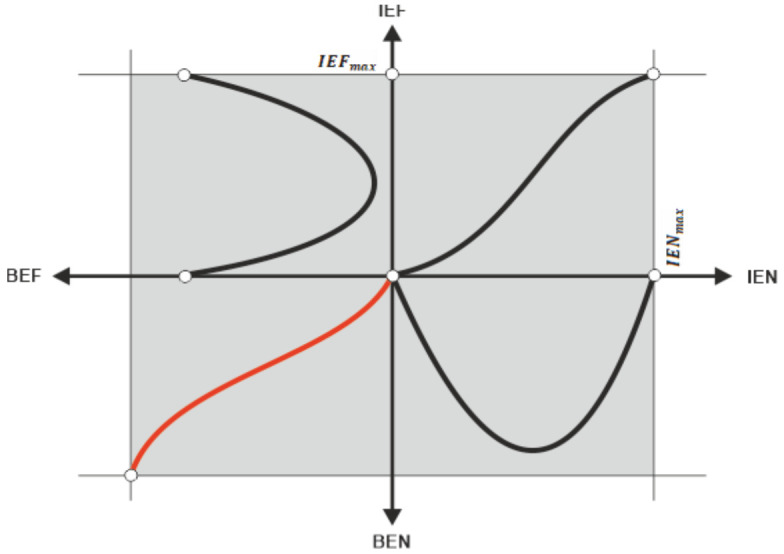
The four-dimensional qualitative analysis of informational and behavioural entropy and efficiency. Source: authors’ graphical construction.

**Figure 5 entropy-23-01396-f005:**
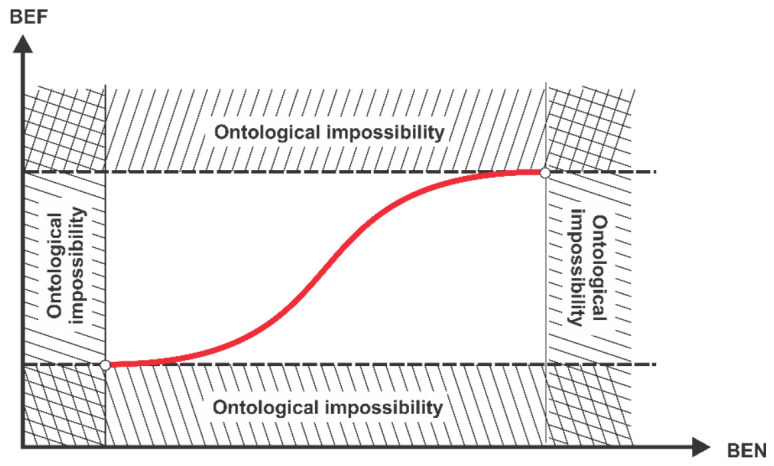
The EBBE logistic curve: $BEF=f\left(BEN\right)$. Source: authors’ graphical construction.

**Figure 6 entropy-23-01396-f006:**
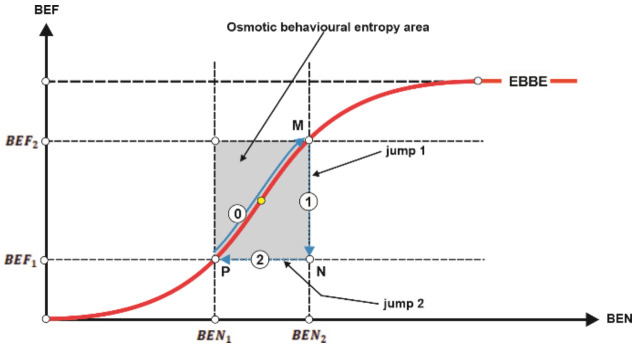
The PABS working. Source: authors’ graphical construction.

**Table 1 entropy-23-01396-t001:** Information processing vs. behaviour processing.

Item	Information Processing	Behaviour Processing
Sphere of information	Any information	Implicit information
Way of processing	Rational	Any
Criterion of stopping	First best (extremizing)	Reachable best (surviving)
Target of processing	Information from information	Information from behaviour

Source: authors’ reasoning.

## Data Availability

Not applicable.

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
