# Peer review of "Entropy-Based Behavioural Efficiency of the Financial Market"

_entropy, 2021, doi:10.3390/e23111396_

Round 1
Reviewer 1 Report
General remarks
The paper is structured properly. Perhaps a classical approach – background, problem, aim, hypotheses, etc. I do not insist but the structure of the paper could be improved.
The idea of the paper is innovative by itself. However, it is not developed properly in the text of the paper.
Assessment of the paper
The paper embodies four problem areas:
- Financial market efficiency
- Information on financial markets.
- Behavioural efficiency of financial markets
- Entropy-based behavioural efficiency.
The first two problem areas show that the Authors are familiar with theory and advanced modelling of financial markets. It can be read from the text.
The proposal of behavioural efficiency stirs a few questions. It is innovative, however, it could be difficult to identify all forms of behavior. It is proposed as a specific extension of informational efficiency. It is described in a very detailed way showing the competences of the Authors but still it looks that the description lacks clarity. I think that another figure with a detailed description of behavioural entropy could be helpful in a better understanding of that concept.
Fig. 1 shows the causal links but leads to the question: how the behaviour will be operationalized?
How the patterns of behaviour presented in Table 1 can be operationalized?
This new concept per se is interesting and perhaps instead of applying a new idea of entropy, the Authors could go into that direction – another paper with more advanced formal modelling?
While the concept of behavioural efficiency is innovative by itself, more doubts concern the idea of behavioural entropy.
The statement does not seem to be so clear: “the behavioural 376 entropy is a result of two interwoven factors, in fact two distinct invisible hands: 1) the 377 micro invisible hand – mIH (namely, the micro-selection), and 2) the macro invisible hand 378 – MIH (namely, the macro-selection). 379
Are truly able to identify the operations of those two “invisible hands”? Perhaps this simplification is excessively crude?
The following questions should be asked (and answered) when proposing the behavioural entropy:
- What are the reasons for developing such a concept? The Authors respond to this question in a detailed way and in and in a very universal way – part 3.1.2 and Figure 3.
I begin to feel uneasy when I see a mixture of philosophical considerations, human behaviour (no matter where), and precisely defined mathematical models. Efficient Market Theory is like that so why do we extend our reasoning to the ontology of ordering on the markets?
Assumptions of non-linearity are quite speculative. Why this shape of the curve? It is explained but is such an approach truly needed.
How to link granulation with the behaviour of market participants.
At this point, we can see that the concept of behavioural entropy demands more explanation concerning its development and usefulness. It looks quite speculative. Perhaps declaring, in the beginning, the problems which can be better understood (solved) with the behavioural entropy in such a way could help to maintain a clear line of reasoning.
Definitely, the ideas illustrated by Fig. 3 seem to be going too far.
Why not concentrate only on the market and not on market creation (emergence)?
- There is another challenge concerning entropy. The Authors present two ideal forms. Of course, informational entropy is relevant to the financial market and the Authors create its specific extension.
However, there are multiple ideas of entropy, and developing behavioural entropy the Authors should expose a better knowledge of entropy. There was a paper in Entropy on the interpretations of entropy in economics written by Aleksander Jakimowicz. As to make it clear, I do not promote that paper but it should be considered in this paper.
I would also suggest checking the works by Philip Mirowski.
Additionally, what is somehow surprising to me, the Authors from Romania do not make any reference to the ideas of Georgescu-Roegen. Just even a short remark.
Conclusions
The paper can be published but the following changes should be considered:
- Clarifying the idea of behavioural entropy and proving that it is just another formula similar to the formulae of entropy but to show that it would help to solve some real problems. It is partly done but insufficiently.
- To show a broader context of discussions in economics and finance about entropy referring to various types of entropy.
- Explaining why broad but in principle simplified, speculative, “ontological” considerations are needed in developing behavioural entropy and linking it with the efficiency of financial market.
- The scope of potential applications of the idea of behavioural entropy is too broad. Either it is a new purely technical idea or something general and speculative. It should be rethought.
- The paper seems to be “conceptually overloaded”. Perhaps the Authors would better manage their undisputable erudition? In my opinion, they are very knowledgeable but in this paper, they just want to say too much.
- Extension of Bibliography. Especially more sources on entropy are needed.
Some small errors in English – style, typing should be corrected.
Reviewer 2 Report
Authors should refer to the literature for definitions of entropy. That is missing in the work. There is also a lack of theoretical and conceptual framework. They are mentioned sporadically in the work . The coherence of work is needed to better understand research.
The literature needs to be better linked to research. To enter more references. The number of references for such serious work should be at least thirty. Especially the latest ones.
The methodology is not well explained. It needs to be expanded and more details entered.
Not so many subheadings are needed in research work. Linking them would improve the structure of the work.
There are no sources below the table and in several figures.
The aim of this research cannot be seen from the abstract. Accordingly, from the conclusion we cannot see whether the work corresponded to the task.
Round 2
Reviewer 2 Report
The text is much better. Explanations were made and new references were entered where requested.
This manuscript is a resubmission of an earlier submission. The following is a list of the peer review reports and author responses from that submission.